# Does peripheral neuroinflammation predict chronicity following whiplash injury? Protocol for a prospective cohort study

Colette Ridehalgh ![ORCID] ,[1,2] Joel Fundaun,[3] Stephen Bremner,[4] Mara Cercignani,[5] Rupert Young,[6] Chetan Trivedy,[7,8] Alex Novak,[9] Jane Greening,[1] Annina Schmid,[3] Andrew Dilley[1]

CR and JF are joint first authors. AS and AD are joint last authors

For numbered affiliations see end of article.

**Correspondence to**
Dr Colette Ridehalgh;
c.ridehalgh@bsms.ac.uk

## ABSTRACT

**Introduction** Whiplash-associated disorder grade 2 (WAD2) is characterised by musculoskeletal pain/tenderness but no apparent nerve injury. However, studies have found clinical features indicative of neuropathy and neuropathic pain. These studies may indicate peripheral nerve inflammation, since preclinical neuritis models found mechanical sensitivity in inflamed, intact nociceptors. The primary aim of this study is to establish the contribution of peripheral neuroinflammation to WAD2 and its role in prognosis. Participants will be invited to participate in a sub-study investigating the contribution of cutaneous small fibre pathology to WAD2.

**Methods and analysis** 115 participants within 1 month following whiplash injury and 34 healthy control participants will be recruited and complete validated questionnaires for pain, function and psychological factors. Data collection will take place at the Universities of Sussex and Oxford, UK. Clinical examination, quantitative sensory testing and blood samples will be undertaken. MRI scans using T2-weighted and diffusion tensor images of the brachial plexus and wrist will determine nerve inflammation and nerve structural changes. Skin biopsies from a substudy will determine structural integrity of dermal and intraepidermal nerve fibres. At 6 months, we will evaluate recovery using Neck Disability Index and a self-rated global recovery question and repeat the outcome measures. Regression analysis will identify differences in MRI parameters, clinical tests and skin biopsies between participants with WAD2 and age/gender-matched controls. Linear and logistic regression analyses will assess if nerve inflammation (MRI parameters) predicts poor outcome. Mixed effects modelling will compare MRI and clinical measures between recovered and non-recovered participants over time.

**Ethics and dissemination** Ethical approval was received from London-Brighton and Sussex Research Ethics Committee (20/PR/0625) and South Central—Oxford C Ethics Committee (18/SC/0263). Written informed consent will be obtained from participants prior to participation in the study. Results will be disseminated through publications in peer-reviewed journals, presentations at national/international conferences and social media.

**Trial registration number** NCT04940923.

### STRENGTHS AND LIMITATIONS OF THIS STUDY

⇒ This study will evaluate the contribution of peripheral nerve inflammation to whiplash-associated disorder grade 2 and its role in prognosis.
⇒ The design will enable us to understand if simple clinical tests can be used to identify nerve inflammation.
⇒ One potential limitation of a two-site study can be variation in data collection between sites, although regular meetings and site visits have been put in place to minimise this.
⇒ The use of different scanners could result in non-homogenous data sets; however, identical scanner types are being used, which should reduce the impact of this potential issue.
⇒ It is not possible to blind the examiner to the status of patients versus healthy control participants, which may influence digital nerve palpation; therefore, care will be taken to ensure similar pressures are applied to both groups where possible.

### INTRODUCTION

Whiplash injury sustained by a motor vehicle collision can cause significant long-term pain and disability. In the UK, economic costs are as high as £3 billion per year mostly due to healthcare costs.[1] Most whiplash cases are considered whiplash-associated disorder grade 2 (WAD2), as defined by the Quebec Task Force Classification system.[2] WAD2 refers to musculoskeletal pain and tenderness but with no frank nerve injury identified during routine clinical testing (eg, electrodiagnostic testing, neurological bedside testing). Approximately 50% of the people with WAD2 develop chronic symptoms that frequently persist for more than 5 years.[3 4] Although a limited number of prognostic factors have been identified,[5] they do not indicate a site of tissue pathology. This lack of understanding of the underlying pathophysiology likely

explains why treatments for this condition are relatively ineffective.[6 7] Despite the absence of a detectable nerve injury on routine clinical testing, many of the clinical features of WAD2 indicate a syndrome that is neuropathic in nature, such as reduced vibration thresholds, cutaneous thermal hypoaesthesia and cutaneous hypersensitivities to tactile and thermal stimuli distal to the site of injury.[3 8–14] Furthermore, clinical tests of nerve trunk mechanical sensitivity, such as palpating over upper limb peripheral nerve trunks and positioning the upper limb in postures that increases the tension in the brachial plexus and median nerve (the upper limb neurodynamic test-1; ULNT1), provoke symptoms in many patients.[8 9 14] Based on these findings, we propose that some people diagnosed with WAD2 may have an undetected nerve pathology, such as peripheral neuroinflammation, which could be a cause of chronicity. Consistent with a role for neuroinflammation, animals with a localised neuritis develop tactile and thermal-evoked cutaneous hypersensitivities in the absence of axonal degeneration.[15–17] Of note, intact nociceptive axons become mechanically sensitive responding to pressure and stretch,[17–20] similar to the neural mechanosensitivity observed in people with WAD2.

Magnetic resonance (MR) neurography has gained increasing interest in the diagnosis of peripheral nerve pathology.[21–23] An increased signal intensity on T2-weighted sequences is considered an imaging correlate of the intraneural oedema associated with peripheral neuroinflammation. Accordingly, T2-weighted signal is increased in animal models of neuroinflammation,[24 25] and correlates positively with histological findings of inflammation.[25] Using T2-weighted MR neurography in a group of patients with chronic WAD2, we have identified increased signal intensity in the cervical nerve roots and the median nerve at the carpal tunnel, indicative of peripheral neuroinflammation.[9] In addition to T2-weighted MR neurography, diffusion tensor imaging (DTI) has recently been employed to examine the microstructural integrity of peripheral nerves.[26 27] DTI evaluates the diffusivity of water molecules within nerves, which can provide information on the integrity of axons and myelin sheaths,[28 29] as well as intraneural oedema.[30]

In addition to neuroinflammation and functional changes to the nervous system, there is also some evidence for the presence of structural nerve pathology in patients with WAD2. This is apparent by a decreased intraepidermal nerve fibre density in the index finger in patients with chronic WAD2.[14] It currently remains unknown whether such small fibre degeneration is already apparent in the early stages after injury and whether it predicts outcome.

The main aim of this study is to establish the contribution of peripheral nerve pathology to WAD2 and its role in prognosis. Our primary focus is on peripheral neuroinflammation, which we will evaluate using a detailed examination, including MRI of the brachial plexus and distal nerve trunks, blood markers and clinical tests of nerve trunk mechanical sensitivity. Additionally, recruited participants will be invited to participate in a substudy that will use skin biopsies to establish the contribution of cutaneous nerve fibre pathology to WAD2. Our results will advance our understanding of the pathology underlying WAD2 and may identify factors that predict the development of persistent symptoms.

## METHODS AND ANALYSIS
### Objectives
The first objective of this study is to determine the presence of peripheral neuroinflammation during the acute stage following a WAD2 injury. This objective will use MRI to assess the extent of peripheral neuroinflammation and nerve microstructure in a cross-section of patients within 4 weeks of injury and healthy controls. The second objective will determine whether the presence of peripheral neuroinflammation in the acute stage can predict patient recovery status at 6 months. The third objective will determine the course of peripheral neuroinflammation from the acute to chronic stages of a WAD2 injury. For this objective, we will examine changes in MRI measures and clinical tests from baseline to 6 months in both recovered and non-recovered patients. The fourth objective will determine whether low-cost clinical tests can serve as markers to detect neuroinflammation as determined on MRI. The objective of the substudy is to determine whether the structural integrity of cutaneous nerve fibres is altered in the acute stage following a WAD2 injury, and whether it contributes to the progression to chronicity.

### Study design
This is a prospective cohort study at two centres. A total of 115 participants will be recruited following a whiplash injury within 1 month of their original injury. We will also recruit 34 healthy proportionally age-matched and gender-matched control participants. People with acute WAD2 and healthy control participants will attend a baseline appointment. In addition to MRI, a detailed musculoskeletal assessment will be performed, as well as tests of nerve trunk mechanical sensitivity and quantitative sensory testing (QST). A blood sample will be taken and analysed for levels of inflammatory markers. Participants will also be asked to complete a series of validated questionnaires to assess neuropathic symptoms, functional deficits, quality of life and psychological factors. For the substudy, skin biopsies from the index finger and ankle will also be taken. At 6 months, the Neck Disability Index and a self-rated global recovery question will be used to determine recovery. A sequential sample of patients will be invited to attend a follow-up appointment during which the same detailed assessments will be repeated (figure 1).

This study has been registered at ClinicalTrials.gov. The protocol version is V2 25 September 2020.

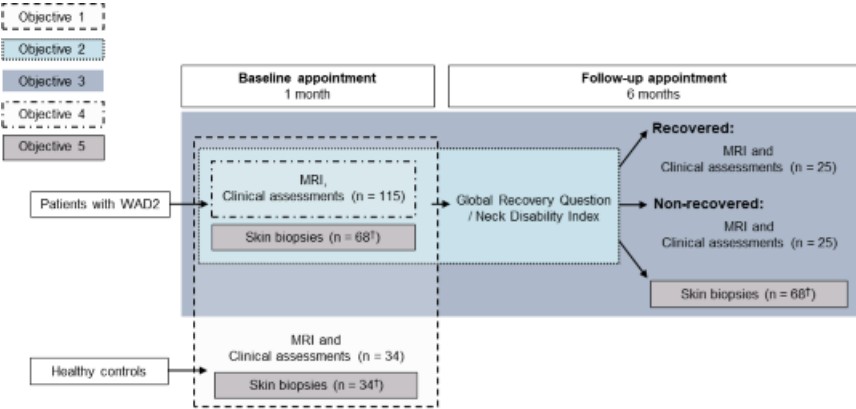

**Figure 1** Flow diagram showing the study design. WAD2, whiplash-associated disorder grade 2.

## Patient and public involvement

Patient partners were involved in the design of the study and highlighted the relevance and need for this study. They will continue to be an integral part of the research team, offering advice from conducting the study through to interpretation of the data and dissemination of findings.

## Participant recruitment

Patients aged 18–60 that meet the Quebec Task Force Classification of WAD2 will be recruited within 4 weeks of injury. Participants will be recruited from emergency departments and minor injury units in Surrey, Sussex, Kent and Oxfordshire, directly by clinicians, and following weekly hospital database searches using defined search terms (eg, motor vehicle collision, muscle injury: neck, bruise/contusion/abrasion: neck, sprain/ligament injury: cervical spine). Potential participants will be contacted in person, or by phone, and asked for permission for their details to be passed on to the study team, and/or sent the study details. In addition, posters will be displayed in departments with study contact details. Healthy age-matched and gender-matched control participants will be recruited through departmental emailing lists (Universities of Sussex and Oxford as well as local hospitals), social media and flyers on public notice boards. Potential participants will be screened for suitability via telephone and also be invited to take part in the substudy. Eligible individuals will be sent a participant information sheet and questionnaires to complete (please see table 1 and questionnaire section below) and will be invited to attend a baseline appointment at either Brighton and Sussex Medical School, University of Sussex, Falmer or the Nuffield Department of Clinical Neurosciences, University of Oxford.

Exclusion criteria include pregnancy, recent history of cervical/arm pain lasting >3 months, previous diagnosis of a peripheral neuropathy, history of systemic illness or autoimmune disease, non-medically controlled hypertension and ongoing steroid treatment. Participants will be assessed for their suitability to undergo MRI. Additionally, healthy control participants must not have had a history of whiplash injury and have not received treatment for neck, thoracic spine or upper limb pain within the past 3 months.

Both recruitment methods and experimental design, in particular length of time of research appointments and MRI scanning, have been discussed and modified based on the comments received from our patient advisory group.

## Baseline and follow-up appointment

Written informed consent will be obtained by a member of the study team at the baseline appointment. Additionally, consent will be gained for use of data/samples in ancillary studies. The following procedures will be carried out (see table 1 for full details):

### Questionnaires

Patients will complete the following validated questionnaires: Neck Disability Index (NDI),[31 32] pain detect questionnaire (painDETECT),[33] short post-traumatic stress inventory (PTSD-8),[34] Pain Catastrophising Scale (PCS),[35] Depression, Anxiety and Stress Scale (DASS42),[36] the quality of life questionnaire (EQ-5D-5L),[37] and the Impact of Events Scale (IES-Revised).[38]

### Clinical assessment

Demographic information and type of neck injury sustained will be obtained (ie, speed of impact, direction of collision and initial symptoms). A body chart and Visual Analogue Scale (0–100 mm) will be completed to describe distribution and severity of current symptoms. Current medication intake will also be recorded. Active range of motion of the cervical spine and upper limbs will be assessed and any pain or restriction in range noted (see table 1 for details). A neurological assessment of the upper extremities will be undertaken of cutaneous hypersensitivity and hypoaesthesia using neurotips and cotton wool, and coins will be used to assess thermal sensitivity. Isometric muscle strength and tendon reflexes will also be assessed (table 1).

### Nerve mechanical sensitivity testing

Test of nerve trunk mechanical sensitivity of the brachial plexus, median and ulnar nerves will be performed

**Table 1** Summary of self-reported and physical assessments

| Test | Description | Type of tool/assessment | Participants |
|---|---|---|---|
| Demographics | Age, gender, height/weight. | Self-reported. | HC and whiplash |
| Injury details | Impact, anticipation. | Self-reported. | Whiplash |
| Medication details | Specific to WAD symptoms. | Self-reported. | Whiplash |
| Symptom history | Description of pain or symptoms. | Self-reported. | Whiplash |
| Pain level | Current pain severity. | Visual Analogue Scale. | Whiplash |
| Management | Any investigations/treatment. | Self-reported. | Whiplash |
| Type of pain | Neuropathic pain. | painDETECT Questionnaire. | Whiplash |
| Disability | Neck disability. | Neck Disability Index. | Whiplash |
| Psychological factors | Post-traumatic stress. Pain catastrophising. Depression, anxiety and stress. Quality of life. | PTSD-8. Pain Catastrophising Scale. DASS 42. EQ-5D-5L. | Whiplash |
| Active range of motion | Cervical spine. Glenohumeral joints. Elbow, wrist and hands. | 0–3 scale (0=full range, no symptoms; 1=full range, symptoms; 2=reduced range, no symptoms; 3=reduced range, symptoms). | HC and whiplash |
| Standardised neurological examination | Sensation (dermatomes C5–T1). Myotomes (C5–T1). Reflexes (biceps, triceps). | Cotton wool (light touch). Neurotip (pinprick). Isometric muscle contraction. Reflex hammer. | HC and whiplash |
| Bedside sensory testing | Sensation—main pain area and index finger (median nerve/C6 innervation). | Warm/cold coin (thermal) and Neurotip (pinprick). | HC and whiplash |
| Quantitative sensory testing—index finger | Warm/cold sensation/pain thresholds. Mechanical detection thresholds. Mechanical pain thresholds. | MSA thermal stimulator (Somedic, Stockholm, Sweden). Von Frey filaments (Marstock NerveTest filaments, MRC Systems, Heidelberg, Germany). Weighted pinprick stimulators (MRC Systems, Heidelberg, Germany). Vibration (Tuning fork). Pressure pain threshold (Algometer, Wagner Force dial, Greenwich, USA). | HC and whiplash |
| Heightened nerve mechanosensitivity | Upper limb neurodynamic test 1 (median nerve bias). Upper limb neurodynamic test 3 (ulnar nerve bias). Nerve palpation. Brachial plexus. Median nerve proximal to carpal tunnel. Ulnar nerve (Guyon's canal). Ulnar nerve (proximal to cubital tunnel). Median nerve (carpal tunnel). Phalens. Tinels. | Positive/negative test (symptom response changes with structural differentiation). Inclinometer (ROM degrees). Positive/negative test (symptom response changes with structural differentiation). Digital palpation (0–3: 0=no pain or discomfort, 1=local discomfort, 2a=local painful response, 2b=referred pain/symptoms). Algometer and score (0–3). Digital palpation (0–1). | HC and whiplash |
| MRI | Brachial plexus. Wrist. | T2-weighted images. Diffusion tensor images. | HC and whiplash |
| Serum samples | Venepuncture. | Inflammatory markers. | HC and whiplash |
| Skin samples | Skin biopsy. | Density of dermal and intraepidermal nerve fibres using immunohistochemistry. | HC and whiplash |

*EQ-5D-5L, Quality of life questionnaire
DASS, Depression, Anxiety and Stress Subscale; HC, healthy controls; PTSD, post-traumatic stress disorder; ROM, range of motion; WAD, whiplash-associated disorder.

bilaterally. The ULNT1 (median nerve bias[39]), will be performed as per standard sequence[40] using 90° abduction and lateral rotation of the shoulder, elbow supination, wrist and finger extension followed by elbow extension. The ULNT3 (ulnar nerve bias[40]) will be performed using shoulder abduction, forearm pronation, wrist and finger extension followed by elbow flexion. The ULNT1 and ULNT3 tests are considered 'positive' when there is reproduction of participant's symptoms and these are changed with the addition of a structural differentiation manoeuvre[39] (table 1) . For the ULNT1, elbow range of flexion at the point of symptom onset will be measured using a digital inclinometer (Trend, DLB Swansea, UK). Mechanosensitivity to pressure will be established by both

response to digital pressure and using an algometer (tip size=1 cm$^2$, Wagner, USA). The algometer will be used over the ulnar nerve at the proximal cubital tunnel and the carpal tunnel bilaterally. Participants will be asked to inform the researcher at the point where the pressure applied over the nerve changes to pain. Measurements will be taken three times at each site and averages used for analysis. Other sites (bilaterally at brachial plexus in the supraclavicular area, median nerve at the proximal wrist crease, ulnar nerve at Guyon's canal) will be palpated digitally and graded according to symptom response (see table 1). Phalen's and Tinel's test will be performed bilaterally at the wrists.[41]

### Quantitative sensory testing

QST will be carried out over the ventral aspect of the proximal phalanx of the index finger and the thenar eminence according to the German Network for Neuropathic Pain protocol.[42] Additionally, we will perform warm detection, cold detection and pressure pain thresholds over the contralateral lower limb (upper anterolateral aspect of the tibia) to assess for any systemic changes to QST. Thermal detection and pain thresholds (heat and cold) will be measured using a Thermotester (Somedic, Sweden) and mechanical detection and pain thresholds with von Frey filaments (Optihair 2 MRC Systems, Germany), Rydel Seiffer tuning fork, an algometer (Wagner, USA) and weighted pinprick stimulators (MRC Systems, Germany). QST will be carried out on the most symptomatic side in WAD2 participants and non-dominant side in healthy control participants.

### MRI)

Images of the brachial plexus and median nerve at the wrist (on the most symptomatic side in patients and dominant side in healthy control participants) will be obtained at each research site using a 3-Tesla scanner of the same manufacturer and model (Siemens Prisma, Siemens Medical Solutions, Erlangen, Germany) with a dedicated 64 channel head/neck coil and a small 4-channel flex coil for the wrist.

For brachial plexus imaging, participants will be positioned supine. Coronal images will be obtained using a two-dimensional multislice T1-weighted (slice thickness=3 mm) and a T2-weighted short tau inversion recovery (STIR) three-dimensional (3D) SPACE sequence (slice thickness=0.8 mm). For the wrist, participants will be positioned with their arm above their head in a reverse superman position with the wrist in the centre of the bore using a flex coil. Axial images will be obtained using a T1-weighted (slice thickness=3.0 mm) and a T2-weighted STIR sequence with flow suppression (slice thickness=1.5 mm). A 3D balanced steady-state free precession (bSSFP) will also be acquired as anatomical reference. The sequence will be repeated twice, with phase cycling, to minimise banding artefacts.[26]

DTI will also be obtained. For the brachial plexus, we will use Readout Segmentation Of Long Variable Echo trains (RESOLVE) sequences,[43] which provide data with low distortions and clinically feasible scan times. For the wrist, we will use diffusion-weighted echo-planar imaging. This acquisition was preferred for the wrist because it provides higher signal-to-noise than RESOLVE.[26] DTI images will be registered to structural images using T1-weighted images for the neck and bSSFP for the wrist.

In the unlikely event of a structural abnormality being noted incidentally, a radiologist or neurologist would be asked to review the scan and, if there remained any concerns, the patient would be contacted through their general practitioner or directly by the neurologist to arrange a formal diagnostic scan.

### Serum collection

Approximately 30 mL of blood will be taken from each participant (BD Vacutainer Tube SST Advance). Serum will be extracted and stored at −80°C in compliance with The Human Tissue Act 2004. Levels of proteins associated with inflammation, such as pro-inflammatory cytokines, will be measured using electrochemiluminescence. All serum samples will be collected and stored in accordance with the Human Tissue Act.

### Sub study: skin biopsies

For the substudy, two 3 mm diameter skin samples will be taken from the ventrolateral aspect of the proximal phalanx of the index finger on the more affected side and 10 cm above the contralateral lateral malleolus under sterile conditions using local anaesthesia (1% lidocaine, 1.0–2.0 mL). We will fix the biopsies in a periodate-lysine-paraformaldehyde solution for 30 min followed by washing samples in 0.1 M phosphate buffer. The tissue will be embedded in an OCT compound and stored at −80°C. Sections of 50 μm will be cut and immunohistochemistry will be performed to visualise cutaneous nerve fibres (eg, with protein gene product 9.5). A single, blinded assessor will evaluate the integrity of intraepidermal and dermal nerve fibres, as previously detailed.[44–46] In brief, intraepidermal nerve fibre density will be counted through the microscope from three samples per patient with the average expressed as fibres per mm epidermis.[46] Dermal innervation will be established by counting the number of dermal nerve bundles per mm$^2$ dermis that include at least five axons.[47] All skin samples will be collected and stored in accordance with the human tissue act.

### Follow-up

All participants will be asked to complete the questionnaires and answer a self-rated global recovery question 6 months after their injury. Twenty five participants who have recovered and 25 who have not recovered will undergo the full protocol. To promote participant retention, a newsletter will be sent to participants to keep them up to date about the study, and a small payment will be given to those who return their completed questionnaires at 6 months or attend their follow-up appointment.

## Data storage and analysis

Data and samples will be anonymously coded, and stored on a secure web platform (REDCap (Research Electronic Data Capture), Vanderbilt University, Nashville, USA) and university freezers, respectively. Anonymised imaging data will be stored in DICOM and NIfTI formats on secure university servers. Participants will be identified by a unique study specific code. All data and sample analyses will be blinded to participant group and time point.

## Image analysis

MRI data will be coded and analysed blindly. T2-weighted signal intensity will be analysed using MATLAB (Mathworks, Natick, USA, release 2022a). Briefly, regions of interest within the brachial plexus and the median nerve will be identified, and a binary mask created to allow for background subtraction. Following background subtraction, the mean pixel grayscale value for each nerve segment will be determined and normalised to a region of adjacent tissue.

Brachial plexus DTI data will be de-noised using a self-supervised machine learning approach[48] implemented in Dipy.[49] Wrist DTI data will be denoised using a method based on random matrix theory,[50] implemented in MRtrix. Subsequently, they will be corrected for susceptibility and eddy current distortions using a combination of TopUp and Eddy (part of the FMRIB's software library). DTI fitting will be performed for both kinds of data using tools from the FMRIB's software library to create fractional anisotropy, mean diffusivity and other parametric maps.[26] Mean DTI parameters will be determined within regions of interest.

## Sample size calculation

Mean nerve signal intensity ratio data from our previous cross-sectional study of 13 controls and 9 participants with chronic WAD2 were analysed using a linear mixed model.[9] The observed difference in means between controls and WAD2 participants was 0.0600 (SE=0.0315) for the C5–C8 roots of the brachial plexus, which was statistically significant; age was not a statistically significant covariate. The intraclass correlation coefficient of the nerve signal ratio was estimated to be 0.45. Assuming that there are eight assessments per participant (two sides × four roots), a SD of 0.1, intraclass correlation coefficient of 0.45 and a ratio of 3 participants with whiplash to one healthy control, the study would require 48 participants with whiplash and 16 controls to detect an effect estimate of 0.06 for the mean nerve signal intensity difference (80% power, 5% significance). Since in the acute stage it is estimated that only 50% of the participants with whiplash will develop chronicity,[51] we will need to image 96 participants with whiplash and 32 controls for objective 1 (difference in neuroinflammation in the acute stage). Based on ~48 participants who have not recovered at 6 months, we would be able to fit a logistic regression model with 5–10 predictor variables for objective 2 (predictive factors of chronicity).

For objective 3 (development of neuroinflammation over time), we will ask 25 participants who have recovered and 25 participants who have not recovered to return at 6 months, which will enable an effect size of 0.9 to be detected using analysis of covariance, with 95% power for 5% significance. This value assumes that the correlation between the acute and chronic measurements is 0.5.

For objective 4 (validity of low cost tests of neuroinflammation), with nine independent variables, a rule of thumb for a multiple linear regression (ie, 10 participants per additional variable) implies that data from 90 participants with whiplash should be adequate to explore this.[52]

For the substudy, 68 participants with whiplash and 34 healthy controls (80% power, 5% significance, 0.53 effect size, one-tailed test) would allow the detection of a 20% smaller difference in intraepidermal nerve fibre density in the acute stage compared with healthy controls based on our previous cohort of patients with chronic WAD2.[53]

Previous longitudinal studies on WAD2 have demonstrated a <15% dropout rate over 6 months,[54 55] and therefore we will recruit 115 participants with acute WAD2 and 34 healthy control participants.

## Statistical analysis

Data will be exported to Stata (V.17, StataCorp, Texas, USA) for analysis. The distribution of the data will be checked for normality, and parametric, or non-parametric methods will be used as appropriate. Participants' characteristics will be described at all time points.

For objective 1, regression analysis, or non-parametric tests, will be used as appropriate to identify differences in MRI parameters (T2-weighted STIR 4 nerve roots per side, nerve morphology and DTI parameters) and clinical tests (eg, neuropathic pain scores, tests of nerve mechanosensitivity, QST and serum inflammatory markers) between participants with acute WAD2 and age/gender-matched controls. Regression analysis will be used to assess associations with age and height.

For objective 2, recovery of participants with WAD2 will be determined at 6 months with the Neck Disability Index and a global recovery question. The analysis will be conducted with three models using different predictors measured in the acute stage: (a) MRI parameters of neuroinflammation, (b) clinical parameters that reflect neuroinflammation and (c) the addition of MRI to clinical parameters to determine whether this improves predictive strength. The dependence between MRI and clinical assessments will be investigated and composite measures added to the model. In addition, a linear regression will be performed of Neck Disability Index scores on measures of neuroinflammation.

For objective 3, mixed effects modelling will be used to compare MRI and clinical measures between recovered and non-recovered participants with WAD2 over time.

For objective 4, multivariable regression will be used to investigate the relationship of each MRI variable with the clinical parameters of neuroinflammation. In addition,

a regression analysis will be performed between clinical measures of neuroinflammation and MRI parameters.

For the substudy, independent t-tests or non-parametric alternatives will be used to compare dermal and intraepidermal nerve fibre density in skin biopsies in patients in the acute phase after whiplash injury with healthy controls. Separate regression models will be used to investigate whether changes in nerve fibre density in the acute stage predict pain persistence at 6 months.

Data reduction techniques (eg, principal components analysis) may be used prior to regression modelling.

Missing data will be managed with multiple imputation strategies.

## Standardisation across the two sites

All clinical test procedures will be piloted between sites to ensure standardisation. Inter-rater reliability of range of motion using the inclinometer for nerve mechanical sensitivity testing will be performed. The German Network of Neuropathic Pain QST protocol has shown to have substantial inter-rater and inter-centre reliability.[56 57] The same MR scanners and coils are available at both sites so that settings will be identical. Any differences between sites will be adjusted at the analysis stage by comparing healthy controls at both sites.

Several quality control checks will be performed including checking of data entry (exchange of data sets between sites) and a comparison of healthy control data obtained at both sites.

## Data sharing

Project data will be made available via open access publications, as well as the study webpage and the Universities of Sussex and Oxford repository, which are openly accessible. A statement will be placed on the study webpage and publications asking new users to contact the principal investigator if they wish to have access to the original data set. Anonymised data will be made accessible on reasonable request once published or following completion of the study, adhering to data protection regulation and the Human Tissue Act (2004).

## Current status of the study

Recruitment of participants to the study commenced in July 2021 and will continue until early summer 2023. The study is expected to be completed by August 2024.

## ETHICS AND DISSEMINATION

The study has received sponsorship from the University of Sussex and ethical approval from the London-Brighton and Sussex Research Ethics Committee (20/PR/0625) and South Central—Oxford C Ethics Committee (18/SC/0263). The study is sponsored by University of Sussex which provides oversight for the totality of research sponsored by the University throughout the life-cycle of the study. Additionally, the study is independently reviewed annually by the funding body. Any protocol amendments will be scrutinised and approved by both the study sponsor and the Health Research Authority and changes to the protocol amended at ClinicalTrials.gov.

The results of this study should be of great interest to people after whiplash injury, clinicians, researchers, legal and insurance companies and policymakers. Dissemination via a multitude of platforms is therefore necessary to enhance its reach. We plan to publish traditionally in open access peer-reviewed journals as well as presenting at national and international conferences. In addition, dissemination will occur via social media outlets. Patient and clinician information resources such as infographics will be produced and disseminated. Additionally, the patient partner group will support best strategies for dissemination to reach patient groups as well as policy makers. Local presentations to stakeholder groups such as patient groups, health authorities and professional bodies will be given.

**Author affiliations**
[1]Department of Neuroscience, Brighton and Sussex Medical School, Brighton, UK
[2]School of Sport and Health Sciences, University of Brighton, Eastbourne, UK
[3]Nuffield Department of Clinical Neurosciences, University of Oxford, Oxford, UK
[4]Department of Primary Care and Public Health, Brighton and Sussex Medical School, Brighton, UK
[5]Cardiff University Brain Research Imaging Centre, Cardiff University, Cardiff, UK
[6]School of engineering and informatics, University of Sussex, Brighton, UK
[7]Emergency Departments, University Hospitals Sussex NHS Foundation Trust, Brighton, UK
[8]Queen Mary University of London, London, UK
[9]Emergency Medicine Research Oxford, Oxford University Hospitals NHS Foundation Trust, Oxford, UK

**Contributors** AD, AS and JG conceptualised the study. CR, JF, JG, AS and AD wrote the manuscript, SB oversaw the statistical aspects of the protocol and MC determined MRI sequence and analyses. All authors (CR, JF, AD, AS, JG, SB, MC, CT, AN, RY) reviewed drafts of the manuscript and approved the final version.

**Funding** This work is supported by a Pain Challenge Grant from Versus Arthritis (number (22465)). AS is supported by a Wellcome Trust Clinical Career Development Fellowship (222101/Z/20/Z) and the Medical Research Foundation (MRF-160-0013-ELP-SCHM-C0842). Her research is supported by the National Institute for Health Research (NIHR) Oxford Biomedical Research Centre (BRC). The views expressed are those of the authors and not necessarily those of the National Health Service, the NIHR or the Department of Health. This research was funded in whole, or in part, by the Wellcome Trust. For the purpose of Open Access, the author has applied a CC BY public copyright licence to any Author Accepted Manuscript version arising from this submission.

**Competing interests** None declared.

**Patient and public involvement** Patients and/or the public were involved in the design, or conduct, or reporting, or dissemination plans of this research. Refer to the Methods section for further details.

**Patient consent for publication** Not applicable.

**Provenance and peer review** Not commissioned; externally peer reviewed.

**ORCID iD**
Colette Ridehalgh http://orcid.org/0000-0003-2226-6531

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
