## [Reviewer comments · BMJ Open]

ARTICLE DETAILS

TITLE (PROVISIONAL)	Does peripheral neuroinflammation predict chronicity following whiplash injury? Protocol for a prospective cohort study
AUTHORS	Ridehalgh, Colette; Fundaun, Joel; Bremner, Stephen; Cercignani, Mara; Young, Rupert; Trivedy, Chetan; Novak, Alex; Greening, Jane; Schmid, Annina; Dille, A.

VERSION 1 – REVIEW

REVIEWER	Jens Astrup University of Copenhagen
REVIEW RETURNED	18-Jul-2022

GENERAL COMMENTS	I favor the ideas behind this study which are to determine a possible degree of neuroinflammation in WAD2 in comparison to HC measured by MRI and skin biopsy, and to relate such inflammation to prognosis. I have one major concern about the design, and that is that analysis of MRI between WAD2 and HC and in WAD2 early and later are not conducted by blinded analysis. Neither is the analysis of the skin biopsies in WAD2 and HC conducted blinded. I fear that this leaves the study wide open to bias. If not the authors should explain clearly why this is not so.
---

REVIEWER	Helge Kasch Danish Pain Research Centre, Dept of Neurology
REVIEW RETURNED	28-Sep-2022

GENERAL COMMENTS	Thanks for the opportunity to review this protocol. It is important to recruit and examine acute WAD participants from early after injury (< 2 weeks) if this should be a subacute prospective follow-up. The HC group should also undergo self-report on pain level/symptoms e.g. Neck pain/headache is abundant and if choosing a group never experiencing neck/arm pain headache etc as control this would probably introduce selection bias. Age/gender/BMI but also education level/ADL/work status could be/are of importance Besides use of steroids also examination of eventually underlying neuropathic conditions e.g. unknown diabetes, TSH, B12- and D-vitamin deficiency, leucocytes, thrombocytes, acute phase reactants, liver enzymes, immunoglobulins, m-component, creatinine. QST tests: pressure algometry not well-described using algometry directly over the the nn ulnaris and medianus. Blinding:
--

	examinations are not performed with blinding, this could introduce bias, e.g. how cautious the examiner will apply pressure on nerves or apply the “Phalen test” and so forth, knowing that controls will be examined at non-dominant side and WAD II at the affected side. However the Baron/Treede/German method (2006) of QST is well-established. Blinding of person evaluating MR scans has not been described. Analysis of skin biopsies: no details of preparation and methods have been mentioned. Blinding not described
--	--

VERSION 1 – AUTHOR RESPONSE

Reviewer 1

6. Analysis of MRI between WAD2 and HC and in WAD2 early and later are not conducted by blinded analysis. Neither is the analysis of the skin biopsies in WAD2 and HC conducted blinded. I fear that this leaves the study wide open to bias. If not the authors should explain clearly why this is not so. Apologies for this omission. Analysis of all data will be conducted blind. In our first draft of the manuscript we indicated that participants were coded (On track changed manuscript Page 12: “Data and samples will be anonymously coded, and stored on a secure web platform (REDCap, Vanderbilt University, Nashville, USA) and university freezers. Anonymised imaging data will be stored in DICOM and NiftI formats on secure university servers. Participants will be identified by a unique study specific code”). However, we were not explicit about blinding and therefore for clarity we have added the following sentence: “ All data and sample analyses will be blinded to participant group and time point” (page 12).

Additionally, under Image Analysis (page 13), the following has been added: “MRI data will be coded and analysed blind.”

Under skin biopsy section we have added “ a single blinded assessor will evaluate the integrity of intraepidermal and dermal fibres” (Page 12 track changed manuscript)

Reviewer 2

7. It is important to recruit and examine acute WAD participants from early after injury (< 2 weeks) if this should be a subacute prospective follow-up.

We agree that ideally participants should be seen as soon as possible after whiplash injury. However, without direct access to patients within ED, and also constraints of access to the MRI scan and participant availability, this is not feasible. Additionally, animal studies suggest that nerve inflammation continues beyond a 2 week period (Bove et al., 2009).

Bove GM, Weissner W, Barbe MF. Long lasting recruitment of immune cells and altered epi-perineurial thickness in focal nerve inflammation induced by complete Freund's adjuvant. J Neuroimmunol. 2009;213(1-2):26-30.

8. Examinations are not performed with blinding, this could introduce bias, e.g. how cautious the examiner will apply pressure on nerves or apply the “Phalen test” and so forth, knowing that controls will be examined at non-dominant side and WAD II at the affected side. However the Baron/Treede/German method (2006) of QST is well-established. Blinding of person evaluating MR scans has not been described.

Please see comment 7 above. All participants will have both limbs examined. The only time that a single limb is examined is for QST (full protocol at the index finger), skin biopsies and MRI (wrist), all other tests are bilateral. Still, even though both sides are assessed, there is the chance that the examiner could tap less hard on the healthy control participant’s wrist compared to a patient participant. We can only offer assurances that both researchers are experienced clinicians (as well as researchers) and have met on a number of occasions to practice protocol and ensure parity of all

techniques. Additionally, we are interested in establishing if pain is referred from the site of palpation, and slowly build up pressure until we reach either the end of resistance of the tissues or we reproduce symptoms. We are aware this is a potential limitation of the study and have listed it as one of the limitations. We are also aware of several published studies that have used nerve palpation in this cohort and have statistically significant differences between groups (see Fundaun et al. 2021), however it could still be argued that such differences could indeed be due to greater pressures being applied to the symptomatic participants.

Fundaun J, Kolski M, Baskozos G, Dilley A, Sterling M, Schmid AB. Nerve pathology and neuropathic pain after whiplash injury: a systematic review and meta-analysis. *Pain*. 2022 Jul 1;163(7):e789-e811. doi: 10.1097/j.pain.0000000000002509

9. The HC group should also undergo self-report on pain level/symptoms e.g. Neck pain/headache is abundant and if choosing a group never experiencing neck/arm pain headache etc as control this would probably introduce selection bias. Age/gender/BMI but also education level/ADL/work status could be/are of importance

Thank you for this comment. Selecting what is a true healthy and comparable control is always challenging, and we agree that recruiting participants without any previous health concerns (particularly musculoskeletal pain) would be difficult. We have instead asked that the participants are free from any of the conditions that we also exclude for the patient group, do not have a history of whiplash and have not received treatment within the past 3 months for neck or upper limb disorders. We are recruiting from the same geographical areas as our patient group. Age and BMI will be examined during our analysis. We absolutely acknowledge that ideally we would be matching multi-dimensions which may impact on pain and health status, and this is a limitation to the study.

We have added the following for clarity “Additionally, healthy control participants must not have had a history of whiplash injury and have not received treatment for neck, thoracic spine or upper limb pain in the past 3 months” (see page 7 track changed document)

10. Besides use of steroids also examination of eventually underlying neuropathic conditions e.g. unknown diabetes, TSH, B12- and D-vitamin deficiency, leucocytes, thrombocytes, acute phase reactants, liver enzymes, immunoglobulins, m-component, creatinine. Identifying unknown conditions is challenging and we have to accept that such conditions could be apparent in our healthy controls as well as our patient group. We ask all participants (healthy controls and patients) if they know of any medical conditions with which they may be living with and if we identified any which we felt could compromise their results from an immune system perspective or indeed influence the nervous system they would be excluded. We do not have the means to run blood tests to check for potentially undiagnosed risk factors of nerve injury.

11. QST tests: pressure algometry not well-described using algometry directly over the the nn ulnaris and medianus.

We are performing standardised QST over the main test site (i.e., thenar eminence), and are using algometry over the nerves in exactly the same way- 3 times over each site. The following has been added within the text (see page 10 of track changed document): “Participants will be asked to inform the researcher at the point where the pressure applied over the nerve changes to pain. Measurements will be taken three times at each site and averages used for analysis.”

12. Analysis of skin biopsies: no details of preparation and methods have been mentioned. Blinding not described

This has been detailed on page 12.

VERSION 2 – REVIEW

REVIEWER	Jens Astrup University of Copenhagen
REVIEW RETURNED	08-Nov-2022

GENERAL COMMENTS	The problem of blinding has been addressed adequately in this revised manuscript
--

REVIEWER	Helge Kasch Danish Pain Research Centre, Dept of Neurology
REVIEW RETURNED	20-Nov-2022

GENERAL COMMENTS	Thanks for the opportunity to re-review the revised paper. Chronic WAD is sometimes seen as being settled/determined/defined after 3 months and in other papers 6 months post-injury, Subacute whiplash injury probably is more likely defined after 14 days to 3 months. It is however difficult to predict future chronicity at a time point already considered as stabilized/chronic. I have no further comments to add.
---